# Bayesian regularization of empirical MDPs

### Abstract

In most applications of model-based Markov decision processes, the parameters for the unknown underlying model are often estimated from the empirical data. Due to noise, the policy learned from the estimated model is often far from the optimal policy of the underlying model. When applied to the environment of the underlying model, the learned policy results in suboptimal performance, thus calling for solutions with better generalization performance. In this work we take a Bayesian perspective and regularize the objective function of the Markov decision process with prior information in order to obtain more robust policies. Two approaches are proposed, one based on $L^1$ regularization and the other on relative entropic regularization. We evaluate our proposed algorithms on synthetic simulations and on real-world search logs of a large scale online shopping store. Our results demonstrate the robustness of regularized MDP policies against the noise present in the models.

## 1 Introduction

A Markov decision process (MDP) is a model $M = (S, A, P, r, \gamma)$. Here $S$ is the discrete state space, with each state represented by $s$. $A$ represents the discrete action space with each action denoted by $a$. $P$ denotes the transition probability tensor where, for each action $a \in A$, $P^a \in \mathbb{R}^{|S| \times |S|}$ is the transition matrix between the states, i.e., $P_{st}^a$ denotes the probability of moving from state $s$ to state $t$ if action $a$ is taken at state $s$. $r$ represents the reward tensor where, for each action $a \in A$, $r^a \in \mathbb{R}^{|S| \times |S|}$ is the reward matrix between the states, i.e., $r_{st}^a$ denotes the reward obtained in moving from state $s$ to state $t$ if action $a$ is taken at state $s$. The discount factor $\gamma \in [0, 1]$ determines the importance given to rewards obtained in the future relative to those collected immediately.

A policy $\pi = (\pi_s^a)$ defines the probabilities of taking action $a$ at state $s$. The goal in the MDP is to find a policy $\pi$ that maximizes the expected discounted cumulative reward $v_s^\pi$ for each state $s$, given by $v_s^\pi \equiv \mathbb{E}_\pi \left[ \sum_{m=0}^\infty \gamma^m r_{s_m, s_{m+1}}^{a_m} | s_0 = s \right]$. In what follows, $S$, $A$, and $\gamma$ are considered to be fixed, and therefore, we often denote the MDP model in short with $M = (P, r)$.

For most applications, the environment is modeled with an unknown underlying MDP $\bar{M} = (\bar{P}, \bar{r})$ that is not directly accessible. The *empirical model* $M = (P, r)$ is often estimated from samples of $\bar{M} = (\bar{P}, \bar{r})$ and an optimal policy

$\pi$ is then learned from $M$. As the two models $M$ and $\bar{M}$ are different due to sampling noise, the policy $\pi$ learned from $M$ is different from the true optimal policy $\bar{\pi}$ of $\bar{M}$. When applying $\pi$ directly to the environment modeled by the underlying MDP $\bar{M}$, one often experiences suboptimal performance.

To give a simple example, consider an MDP with only two actions $a_1$ and $a_2$ at each state and that action $a_1$ is always better than $a_2$ for all states $s \in S$ under the true transition probability matrices $\bar{P}^a$ and reward matrices $\bar{r}^a$. The transition matrices $P^a$ and reward matrices $r^a$ constructed from samples are different from $\bar{P}^a$ and $\bar{r}^a$ due to the noise present in the data. As a result, the policy $\pi$ learned from the empirical model $M = (P, r)$ may recommend action $a_2$ over $a_1$ for some states $s \in S$, thus leading to poor generalization performance.

As a more concrete example, let us consider the shopping experience of a customer at an online shopping store. A customer starts a session by typing in an initial query. Based on the given query, the store can recommend products using one of the existing search algorithms. Upon viewing the results, the customer may either make a purchase or continue the browsing session by typing a modified or new query. Such a shopping experience can be modeled as an MDP, where each query is regarded as a state $s \in S$ and each search algorithm as an action $a \in A$. The reward $r_{s,t}^a$ corresponds for example to whether a purchase is made or not. The optimization problem is to decide which search algorithm should be activated given the query in order to improve the overall shopping experience. In a typical offline learning setting, the empirical MDP model $M$ is constructed based on the historical log data. Therefore, the transition tensor $P$ and purchase actions are inherently noisy. If one learns the policy directly from the noisy empirical model, it can have a poor performance when deployed in the future. While we motivate this issue using the online shopping example, it exists universally in many other applications.

In this work, we study the problem of learning a robust MDP policy from the empirical model $M$ that can perform significantly better than the naive policy $\pi$ from $M$ when deployed to the unknown underlying $\bar{M}$. Though this is a challenging problem stated as it is, a key observation is that in many real applications there is often prior information on the rankings of the available actions. Here, we take a Bayesian approach to incorporate such prior information as a

regularizer.

**Main Contributions.** The main contributions of this work are:

- We propose a Bayesian approach that factors in known prior information about the actions and learns policies robust to noise in empirical MDPs. More specifically, two approaches are proposed: one based on $L^1$ regularization and the other on relative entropic regularization. Both can be implemented efficiently by leveraging existing algorithms for MDP optimization.

- We evaluate the designed algorithms on both synthetic simulations and on the logs of a real-world online shopping store dataset. Our regularized policies significantly outperform the un-regularized MDP policies.

**Related work.** When solving the MDPs, entropy regularization has proven quite useful (Peters, Mulling, and Altun 2010; Fox, Pakman, and Tishby 2015; Schulman et al. 2015; Mnih et al. 2016). Commonly, Shannon entropy or negative conditional entropy is used to regularize the MDPs (Peters, Mulling, and Altun 2010; Fox, Pakman, and Tishby 2015; Schulman et al. 2015; Mnih et al. 2016; Dai et al. 2018; Haarnoja et al. 2018). While this results in more robust stochastic policies, they do not necessarily account for any prior information. The work of (Neu, Jonsson, and Gómez 2017; Peters, Mulling, and Altun 2010; Nachum et al. 2017; Wu, Tucker, and Nachum 2019) discusses relative entropic regularization in MDPs, which biases results to a reference distribution. These works focus on improving the convergence and stability of RL methods by employing entropic regularization. But, this idea has yet to be applied in the context of empirical MDPs through a Bayesian perspective.

There has been work on reward shaping (Ng, Harada, and Russell 1999; Harutyunyan et al. 2015; Cooper and Rangarajan 2012; Grzes 2017; Gimelfarb, Sanner, and Lee 2018) where the idea is to obtain a new MDP model $M' = (P, r')$ by modifying the rewards of model $M = (P, r)$ as $r'^a_{s,t} = r^a_{s,t} + \phi(s) - \phi(t)$, where $\phi(s), \phi(t)$ are potential functions at state $s, t$. In particular, (Ng, Harada, and Russell 1999) showed that such a reward shaping ensures that the optimal policy in the two models $M$ and $M'$ remains the same. The focus of all these works is to design potential functions $\phi$ to improve the convergence of algorithms in $M'$ without altering the optimal policies. As the empirical model $M$ and true model $\bar{M}$ are different in our setting due to the inherent noise, we need regularization based approaches which incorporate the prior information about preference towards certain actions as the optimal policy in $M$ is not necessarily optimal under $\bar{M}$.

An alternative solution to this problem would be from a denoising perspective, where the empirical model $M$ is first denoised and then the policy is learned from the denoised model instead. This has been studied in the context of linear systems where the objective is to solve the system $\bar{A}x = \bar{b}$ but the estimated model parameters $A \approx \bar{A}$ and $b \approx \bar{b}$ contain a significant level of noise. To tackle this, (Etter and Ying 2020, 2021) propose an operator augmentation approach that perturbs the inverse of the sampled operator $A^{-1}$ for better

approximation to $x$. However, it is not clear how to extend this approach to the control setting in MDP.

A closely related line of work is that of model based Bayesian reinforcement learning (Ghavamzadeh et al. 2016), where priors are expressed over model as opposed to the policy. Imposing priors in such a way allows one to deal with imprecise models (Levine et al. 2020). Our work on studying Bayesian regularization policies in the action space is complementary to this line of work. The choice of imposing a prior on model against a prior on policy boils down to the application domain. In several application domains, the state space is quite large as a result of which working with a Bayesian model on transition/reward tensors becomes infeasible. In contrast, the action space is relatively much smaller, as a result of which employing a Bayesian approach on the action space is much more practical.

**Organization.** In Section 2, we present the optimization formulations and describe our regularization approach by incorporating prior information. Section 3 studies the performance of the proposed policy against baseline algorithms on several simulated examples. Section 4 evaluates the performance of our proposed algorithms on an application data set.

## 2 Problem statement and algorithms

### 2.1 Policy maximization

Let $\Delta = \{\eta = (\eta^a)_{a \in A} : \sum_{a \in A} \eta^a = 1 \text{ and } \eta^a \geq 0\}$ be the probability simplex over the action set $A$. The set of all valid policies is

$$\Delta^{|S|} = \{\pi = (\pi_s)_{s \in S} : \pi_s \in \Delta \text{ for } \forall s \in S\}.$$

For a policy $\pi \in \Delta^{|S|}$, the transition matrix $P^\pi \in \mathbb{R}^{|S| \times |S|}$ under the policy $\pi$ is defined as $P^\pi_{st} = \sum_{a \in A} P^a_{st} \pi^a_s$, i.e., $P^\pi_{st}$ is the probability of arriving at state $t$ from state $s$ if policy $\pi$ is taken. Similarly, the reward $r^\pi \in \mathbb{R}^{|S|}$ under the policy $\pi$ is given by $r^\pi_s = \sum_{a \in A} r^a_s \pi^a_s$, where $r^a_s = \sum_{t \in S} r^a_{st} P^a_{st}$, i.e., the expected reward at state $s$ under action $a$.

For a discounted MDP (Sutton and Barto 2018; Puterman 2014) with $\gamma \in [0, 1]$, the value function under policy $\pi$ is a vector $v^\pi \in \mathbb{R}^{|S|}$, where each entry $v^\pi_s$ represents the expected discounted cumulative reward starting from state $s$ under the policy $\pi$, i.e.,

$$v^\pi_s = \mathbb{E}\left[\sum_{m=0}^{\infty} \gamma^m r^{a_m}_{s_m, s_{m+1}} | s_0 = s\right],$$

with the expectation taken over $a_m \sim \pi_{s_m}$ and $s_{m+1} \sim P^{a_m}_{s_m,\cdot}$ for all $m \geq 0$. The value function satisfies the Bellman equation (Bellman 1966), i.e., for any $s \in S$

$$v^\pi_s = r^\pi_s + \gamma \mathbb{E}^\pi[v^\pi_{s_1} | s_0 = s] = r^\pi_s + \gamma \sum_{t \in S} P^\pi_{st} v^\pi_t,$$

or equivalently in the matrix-vector notation

$$v^\pi = r^\pi + \gamma P^\pi v^\pi \iff v^\pi = (I - \gamma P^\pi)^{-1} r^\pi.$$

Here, the inverse of the matrix $(I - \gamma P^\pi)$ exists whenever $\gamma < 1$ or there exists a terminal state $z$ in the MDP such

that $P^a_{z,z} = 1$, $r^a_{z,z} = 0$, and $P^a_{s,z} \neq 0$ $\forall a \in A, s \in S$ (Bell 1965). Given the MDP, the optimization problem is

$$\max_\pi \; e^\intercal v^\pi = \max_\pi e^\intercal (I - \gamma P^\pi)^{-1} r^\pi, \qquad (1)$$

where $e \in \mathbb{R}^{|S|}$ is an arbitrary vector with positive entries (Ye 2011). By introducing the discounted visitation count $w^\pi = (I - \gamma P^\pi)^{-\intercal} e$, we can rewrite Equation (1) as

$$\max_\pi \; (w^\pi)^\intercal r^\pi, \quad \text{with} \quad w^\pi \equiv (I - \gamma P^\pi)^{-\intercal} e. \quad (2)$$

## 2.2 Bayesian approaches

Given an MDP model $M$, we can view (2) as the maximum a posteriori probability (MAP) estimate $\max_\pi \Pr(\pi|M)$ with

$$\Pr(\pi|M) \propto \exp(e^\intercal v^\pi) = \exp((w^\pi)^\intercal r^\pi). \qquad (3)$$

When prior knowledge about $\pi$ is not available, it is natural to take a uniform prior over $\pi$, i.e. $\Pr(\pi)$ is constant. This implies that $\Pr(\pi|M) \propto \Pr(M|\pi) \Pr(\pi) \propto \Pr(M|\pi)$, leading to

$$\Pr(M|\pi) \propto \exp((w^\pi)^\intercal r^\pi). \qquad (4)$$

On the other hand, if prior knowledge about $\pi$ is available, it makes sense to impose more informative priors on $\pi$. One commonly used prior is that at state $s \in S$ an action $\xi(s) \in A$ is often preferred over the rest of the actions. This prior information can be incorporated naturally in the following two ways.

### 2.2.1 $L^1$-type prior

In particular, we assume a prior

$$\Pr(\pi) \propto \exp(-\lambda(w^\pi)^\intercal f^\pi), \qquad (5)$$

where $f^\pi$ is defined to be the $L^1$ norm of $\pi$ outside of action $\xi(s)$, i.e., $(f^\pi)_s = \sum_{a \neq \xi(s)} \pi_{s,a}$. This prior puts more probability on action $\xi(s)$ relative to other actions in $A$. Combining (4) and (5) leads to the a posteriori probability

$$\Pr(\pi|M) \propto \Pr(\pi, M) = \Pr(M|\pi) \Pr(\pi)$$
$$\propto \exp((w^\pi)^\intercal r^\pi) \exp(-\lambda(w^\pi)^\intercal f^\pi).$$

The corresponding MAP estimate is

$$\arg \max_\pi \; \exp((w^\pi)^\intercal r^\pi - \lambda(w^\pi)^\intercal f^\pi)$$
$$= \arg \max_\pi \; (w^\pi)^\intercal (r^\pi - \lambda f^\pi) \qquad (6)$$

Note that individual component of $r^\pi - \lambda f^\pi$ can be broken down as,

$$r^\pi_s - \lambda f^\pi_s = \sum_a r^a_s \pi^a_s - \lambda \sum_{a \neq \xi(s)} \pi^a_s = \sum_a (r^a_s - \lambda \delta_{a \neq \xi(s)}) \pi^a_s,$$

where $\delta_{a \neq \xi(s)} = 1$ if $a \neq \xi(s)$ and 0 if $a = \xi(s)$. Hence, this formulation is equivalent to replacing $r^a_s$ with $r^a_s - \lambda \delta_{a \neq \xi(s)}$, i.e., the reward of all non-preferred actions $a \neq \xi(s)$ is reduced by a constant $\lambda$. The corresponding Bellman equation is

$$v_s = \max_a ((r^a_s - \lambda \delta_{a \neq \xi(s)}) + \gamma (P^a v)_s).$$

Once $v_s$ is computed, the optimal action at state $s$ is given by

$$\arg \max_a ((r^a_s - \lambda \delta_{a \neq \xi(s)}) + \gamma (P^a v)_s). \qquad (7)$$

### 2.2.2 Relative entropy regularization

In the MDP literature, it is common to use Shannon entropy regularization, which allows for learning stochastic policies instead of deterministic ones. However, it fails to capture the prior information, such as the scenario where one of the actions in $A$ is preferred over others. To accommodate such a prior, we propose to use the relative entropy instead of Shannon entropy. By choosing the prior distribution carefully, relative to which the entropy of policy is evaluated, we obtain solutions that prefer one action over other actions in $A$. We consider a penalty such that $\Pr(\pi) \propto \exp(-\kappa(w^\pi)^\intercal h^\pi)$ with $h^\pi_s = \sum_a \pi^a_s \log\left(\frac{\pi^a_s}{q^a_s}\right)$, where $q^a_s$ is a distribution over $A$ that prefers $a = \xi(s)$, e.g.,

$$q_a = \begin{cases} 1 - \epsilon, & a = \xi(s) \\ \epsilon/(|A| - 1), & a \neq \xi(s), \end{cases}$$

for some $\epsilon > 0$. The corresponding MAP estimate is

$$\arg \max_\pi \; (w^\pi)^\intercal r^\pi - \kappa(w^\pi)^\intercal h^\pi$$
$$= \arg \max_\pi \sum_s w^\pi_s \left( r^\pi_s - \kappa \sum_a \pi^a_s \log \frac{\pi^a_s}{q^a_s} \right)$$
$$= \arg \max_\pi \sum_s w^\pi_s \left( \sum_a ((r^a_s + \kappa \log q^a_s) - \kappa \log \pi^a_s) \pi^a_s \right) \qquad (8)$$

This is in fact equivalent to the standard Shannon entropy regularization with modified rewards $r^a_s + \kappa \log q^a_s$, i.e., a penalty $\kappa \log q^a_s$ is added to $r^a_s$ when action $a \in A$ is taken. The magnitude of the penalty for action $a$ is large if the prior probability $q^a_s$ of selecting action $a$ is small. By applying the Bellman equation of Shannon entropy regularization (Neu, Jonsson, and Gómez 2017; Ying and Zhu 2020) to (8), we obtain

$$v_s = \max_{\pi_s \in \Delta} \sum_{a \in A} (r^a_s + \kappa \log q^a_s + \gamma \sum_{t \in S} P^a_{st} v_t - \kappa \log \pi^a_s) \pi^a_s.$$

Because of the Gibbs variational principle, the RHS is equal to $\kappa \log \left( \sum_{a \in A} \exp \left( \frac{r^a_s + \kappa \log q^a_s + \gamma \sum_{t \in S} P^a_{st} v_t}{\kappa} \right) \right)$. Thus, the Bellman equation can be written in the following log-sum-exp form

$$v_s = \kappa \log \left( \sum_{a \in A} \exp \left( \frac{r^a_s + \kappa \log q^a_s + \gamma \sum_{t \in S} P^a_{st} v_t}{\kappa} \right) \right), \qquad (9)$$

which can be solved with a value function iteration. Once $v_s$ is known, the optimal policy at state $s$ and action $a$ is given by

$$\frac{\exp(r^a_s + \kappa \log q^a_s + \gamma \sum_{t \in S} P^a_{st} v_t - v_s)}{Z_s}, \qquad (10)$$

where $Z_s = \sum_{a \in A} \exp(r^a_s + \kappa \log q^a_s + \gamma \sum_{t \in S} P^a_{st} v_t - v_s)$ is the normalization factor.

**Comments** Both the optimization formulations in Sections 2.2.1 and 2.2.2 add a penalty on top of the reward obtained for each action. The less preferred actions (i.e., $a \neq \xi(s)$)

are penalized and hence as a result the learned policy prefers action $\xi(s)$ over the other ones. The magnitude of the penalty depends on the regularization parameter $\lambda$ in the $L^1$ case and $(\kappa, q^a)$ in the relative entropy case. The policy obtained in Section 2.2.1 is a deterministic policy, whereas the one learned in Section 2.2.2 is a stochastic policy due to the added entropy regularization. When $\kappa$ in Section 2.2.2 is chosen to be small, the policy becomes more and more concentrated and is often practically equivalent to the one in Section 2.2.1. Finally, we note that both the approaches presented above can be easily extended to settings where a certain subset of the actions are preferred over the others.

# 3 Simulated examples

In the following simulated examples, we demonstrate numerically that an optimal policy of the empirical MDP $M$ results in sub-optimal performance on the underlying MDP model $\bar{M}$ and that the regularized policies provide significantly better performance.

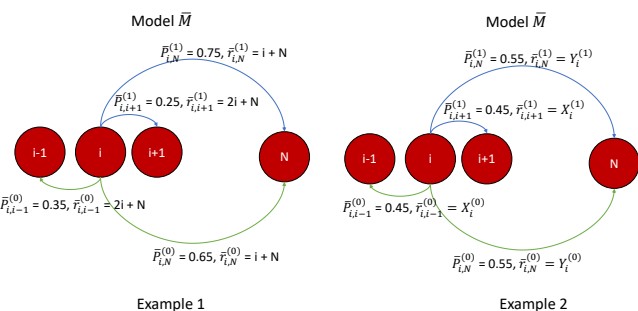

Figure 1: The transition probabilities $\bar{P}^a_{st}$ and rewards $\bar{r}^a_{st}$ of the true model $\bar{M}$ in Example 1 (left) and Example 2 (right). The state space is the set $\{1, 2, 3, \ldots, N\}$. Under action 0, for $i \neq 1$ the transition from state $i$ can go to state $i-1$ or $N$, while for $i = 1$ it can go to $N - 1$ or $N$. Similarly under action 1, for $i \neq N - 1$ the transition from state $i$ can go to $i + 1$ or $N$, while for $i = N - 1$ it can go to state 1 or $N$.

**Example 1.** Consider the MDP model $\bar{M}$ shown in Figure 1. This model has $N$ states $\{1, 2, \ldots, N\}$ and two actions $\{0, 1\}$. The transition and reward tensors for model $\bar{M}$ are defined below

$$
\begin{cases}
\bar{P}^{(0)}_{i,N-1-(N-i)\%(N-1)} = 0.35, & \bar{P}^{(0)}_{i,N} = 0.65 \\
\bar{r}^{(0)}_{i,N-1-(N-i)\%(N-1)} = 2i + N, & \bar{r}^{(0)}_{i,N} = i + N \\
\bar{P}^{(1)}_{i,i\%(N-1)+1} = 0.25, & \bar{P}^{(1)}_{i,N} = 0.75 \\
\bar{r}^{(1)}_{i,i\%(N-1)+1} = 2i + N, & \bar{r}^{(1)}_{i,N} = i + N.
\end{cases}
$$

Here, state $N$ is a terminal state, where $\bar{P}^{0/1}_{N,N} = 1$ and $\bar{r}^{0/1}_{N,N} = 0$. In our simulations, $N = 10$ and $\gamma = 1$. The empirical MDP $M$ is constructed from the true MDP $\bar{M}$ by sampling the transition probabilities from a set of 100 samples for each state $s \in S$. Due to the sampling noise, the

transition tensor $P$ of model $M$ is different from $\bar{P}$ of $\bar{M}$. In this example, we assume that the underlying reward tensor $\bar{r}$ is known exactly. Under the true model $\bar{M}$, action $\xi(s) \equiv 0$ is optimal for any state $s \in S$, as the transition probability to states with higher reward is larger under action 0. However, when the model $M$ is constructed from empirical samples, the optimal policy learned on $M$ recommends action 1 for some states, leading to a sub-optimal performance on the true model $\bar{M}$.

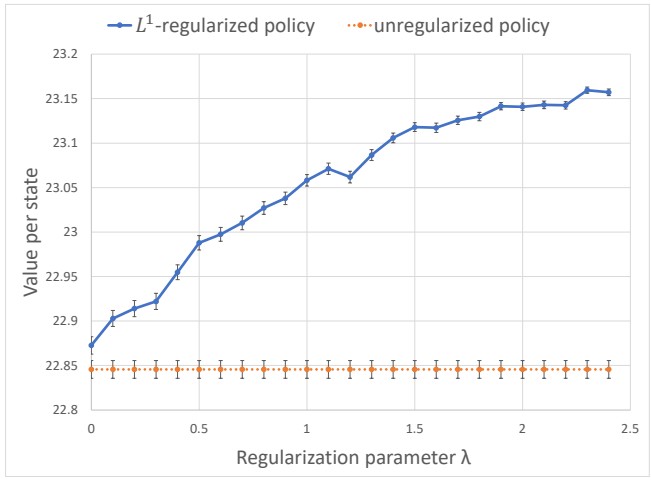

Figure 2: Example 1. The $L^1$-regularized policy vs. the unregularized policy. When $\lambda$ increases, the $L^1$-regularized policy prefers action 0 over action 1 and achieves significantly better value per state on the true model $\bar{M}$ relative to the unregularized policy of model $M$. The error bars indicate a width of two standard error in all subsequent simulations and experiments.

Figure 2 shows the comparison between the $L^1$-regularized policy and the unregularized policy when evaluated on $\bar{M}$. As the regularization parameter $\lambda$ increases, the learned policy prefers action 0 over action 1 and obtains higher value per state as a result. In Figure 3, we compare the relative entropic regularization policy (referred in short as the *RE-regularized policy*) with the unregularized policy on the true model $\bar{M}$. We set regularization coefficient to be $\kappa = 0.25$ and vary the value of prior $q^1_s$ on the $x$-axis. The value $q^1_s = 0.5$ corresponds to the case with Shannon entropic regularization. As the value of $q^1_s$ becomes smaller, the RE-regularized policy prefers action 0 over action 1 and results in a higher value per state relative to the unregularized policy.

In practice, the values of $(\kappa, q^1_s)$ (for the RE-regularized policy) and $\lambda$ (for the $L^1$-regularized policy) can be learned through evaluation on a validation set. In both the cases, the need for regularization goes down as the number of samples used to evaluate $\bar{M}$ increases. This effect is demonstrated in Figure 4, where we plot the performance of the $L^1$-regularized policy as a function of the samples used to estimate the transition probabilities for each state in $\bar{M}$.

**Example 2.** In Example 1, only the transition tensor in $M$ is sampled from the true model $\bar{M}$. In practice, the reward

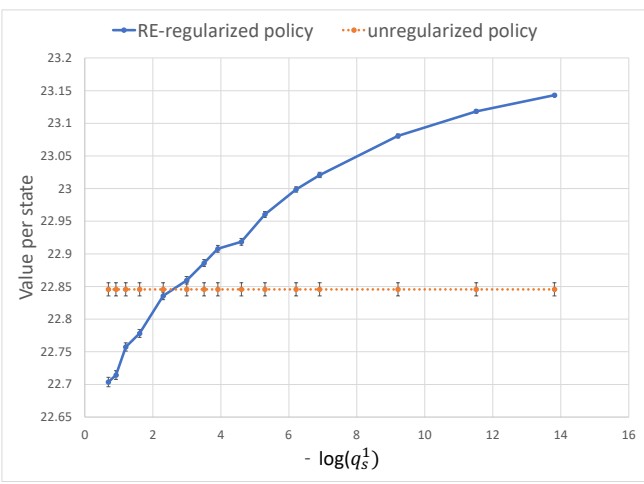

Figure 3: Example 1. The RE-regularized policy vs. the unregularized policy. As the value of $q_s^1$ decreases (or equivalently $-\log(q_s^1)$ increases), the RE-regularized policy prefers action 0 over action 1, leading to improvement in performance over the unregularized policy.

tensor is also estimated empirically. To model this, in this example the reward values $r_{st}^a$ has a Gaussian noise $\mathcal{N}(0, \sigma^2)$ added to the true unknown underlying reward $\bar{r}_{st}^a$. The transition and reward tensors for model $\bar{M}$ are defined below (also see Figure 1)

$$\begin{cases} \bar{P}_{i,N-1-(N-i)\%(N-1)}^{(0)} = 0.45, & \bar{P}_{i,N}^{(0)} = 0.55 \\ \bar{r}_{i,N-1-(N-i)\%(N-1)}^{(0)} = X_i^{(0)}, & \bar{r}_{i,N}^{(0)} = Y_i^{(0)} \\ \bar{P}_{i,i\%(N-1)+1}^{(1)} = 0.45, & \bar{P}_{i,N}^{(1)} = 0.55 \\ \bar{r}_{i,i\%(N-1)+1}^{(1)} = X_i^{(1)}, & \bar{r}_{i,N}^{(1)} = Y_i^{(1)}. \end{cases}$$

Here, $X_i^{(0)}$ is taken to be a random realization drawn from $\mathcal{N}(5, 1)$. Similarly, $X_i^{(1)} \sim \mathcal{N}(6, 1)$, $Y_i^{(0)} \sim \mathcal{N}(2, 1)$, $Y_i^{(1)} \sim \mathcal{N}(3, 1)$. In our simulations, $N = 1000$. For this MDP model $\bar{M}$, the optimal action is 0 for about 82.4% of the states in $\bar{M}$ as the expected number of steps to reach the terminal state $N$ from a given state $s$ is higher in action 0 relative to action 1.

The empirical MDP $M$ is obtained by averaging 100 samples per state of $\bar{M}$, where each reward entry of $r$ is corrupted by a zero mean Gaussian noise with standard deviation $\sigma = 1.5$. As the transition and reward tensors in $M$ contain noise, the unregularized policy from $M$ recommends action 1 for more than 30% of the states. As a result, the unregularized policy is sub-optimal on the underlying model $\bar{M}$.

The prior information that action 0 is preferred over action 1 helps the $L^1$-regularized policy and RE-regularized policy to outperform the unregularized policy. Figures 5 and 6 illustrate this improvement as a function of regularization parameters $\lambda$ and $q_s^1$ (the regularization coefficient $\kappa$ is fixed at 0.25 for this example). As the value of $\lambda$ or $-\log(q_s^1)$ increase, the regularized policies favor action 0 over action

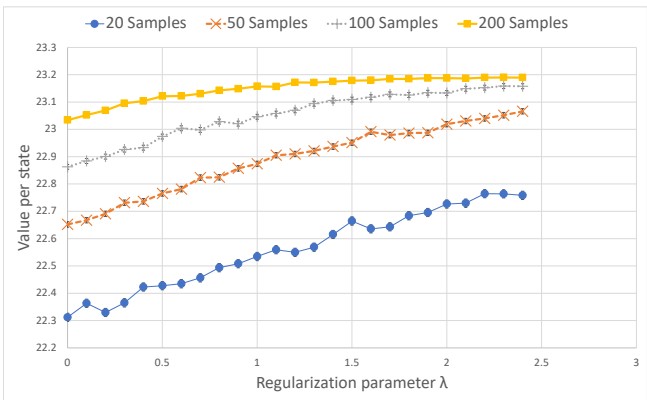

Figure 4: Example 1. The performance of the $L^1$-regularized policy as a function of the samples used to estimate the transition probabilities for each state in $\bar{M}$. The regularization benefit decreases as the number of samples increases due to a reduction of the sampling noise.

1, leading to significant improvements on model $\bar{M}$. With further increase in $\lambda$ and $-\log(q_s^1)$, the performance dips afterwards as the regularized policies start selecting action 0 over action 1 for more states than necessary.

## 4 Experiments on real data

This section discusses large-scale experiments on logs of an online shopping store with competing search algorithms. We consider the user shopping experience discussed in Section 1 and model a shopping session with an MDP where a state $s \in S$ corresponds to the search query typed in by the user.

For each query $s$, the shopping store needs to decide on the search algorithm to use to display results. This corresponds to the two actions of the MDP. When an action is taken, the search results are shown to the user and the user interaction will result in a transition to a new state. We identify this new state with the new query $t$ from the user. However, before making this transition from state $s$ to $t$, the user may make a purchase, which corresponds to the reward. The user may terminate the session at any point with/without making a purchase and this is captured by the transition to a terminal state (see Figure 7). The rewards are considered to be binary: if a user makes a purchase at state $s$ under shopping store's action $a$ and then transitions to $t$, $r_{s,t}^a = 1$. $r_{s,t}^a = 0$ if no purchase were made. The two available search algorithms perform differently on different queries. Therefore, there is an opportunity to interleave different algorithms based on the queries, even within a single shopping session. Moreover, often it is known a priori that one search algorithm may work better than the other. As a result, it is useful to incorporate this information as a prior and design regularized policies that are robust to noise in the empirical MDP.

To conduct our experiment, we collected the search logs of an online shopping store for a time period for two different search algorithms, one deployed in period 1 and the other in period 2, with period 1 is before period 2 in time and both the time periods are non-overlapping. There-

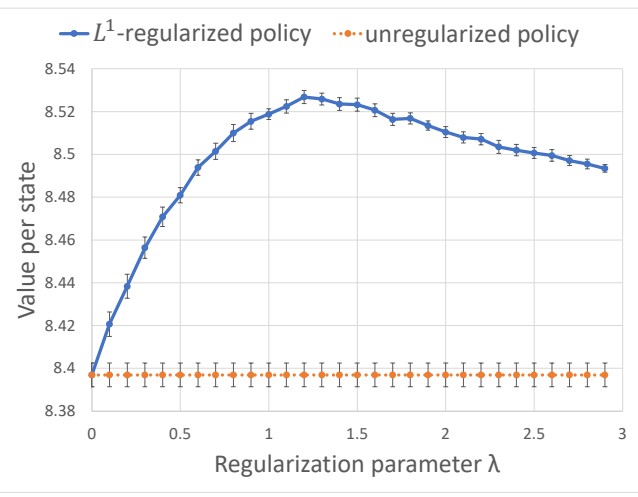

Figure 5: Example 2. The $L^1$-regularized policy vs. the unregularized policy. As $\lambda$ increases, the $L^1$-regularized policy favors action 0 over action 1. The incorporated prior information allows the $L^1$-regularized policy to outperform the unregularized policy.

fore, the action space consists of two search algorithms, $A = \{ranker1, ranker2\}$ which were previously deployed in period 1 and period 2 respectively. We processed the data in a time period where ranker 1 was deployed to obtain the search logs under the ranker 1 action. Similarly, the data for another time period, in which ranker 2 was used, was collected to obtain the user logs under the ranker 2 action. We considered the set of $135,000$ most typed queries as the state space $S$. For each of these time periods, we estimated the transition and reward tensors from user logs, thereby obtaining the MDP model $M$, i.e., $P_{s,t}^{algo1}$, $P_{s,t}^{algo2}$, $r_{s,t}^{algo1}$, $r_{s,t}^{algo2}$ for all $s, t$.

The key challenge is to learn robust policies from the empirical model $M$. It is known a priori that on average, ranker 2 tends to produce better results relative to ranker 1. We exploit this information to learn the $L^1$-regularized and RE-regularized policies, which interleave ranker 1 and ranker 2 effectively for different queries within a single session.

The performance of different policies is judged based on the objective function $e^\intercal v \equiv \sum_{s \in S} e_s v_s$, where $v_s$ is the value function at $s \in S$ with discount factor $\gamma = 1$ and $e_s$ denotes the probability that $s$ is the first query in a random shopping session. This probability $\{e_s\}$ is evaluated based on a hold-out time period from the collected search logs.

In order to evaluate the performance in an unbiased way, we extracted the data for different time periods in which ranker 1 and ranker 2 were deployed, to construct a model $\tilde{M}$ with tensors denoted by $\tilde{P}$ and $\tilde{r}$. As the true underlying model $\bar{M}$ is not directly accessible, we use $\tilde{M}$, a fresh unbiased estimator of $\bar{M}$, to evaluate the different policies. This essentially corresponds to evaluating the performance of the policies on a new time period.

The transition and reward tensors between models $M =$

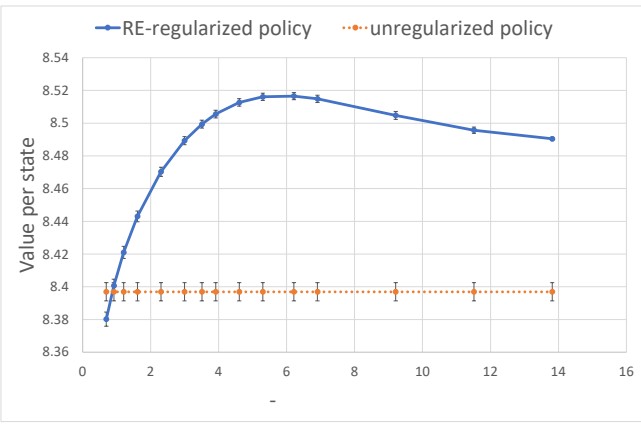

Figure 6: Example 2. The RE-regularized policy vs. the unregularized policy. As $q_s^1$ decreases, the value of $-\log(q_s^1)$ increases and the RE-regularized policy favors action 0 over action 1. Accounting for prior information through $q_s^1$ helps the RE-regularized policy to outperform the unregularized policy.

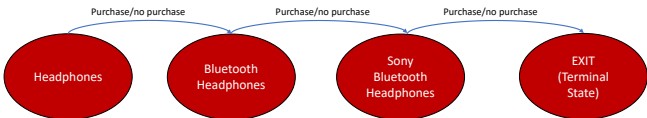

Figure 7: A shopping session at an online shopping store is typically sequential in nature. The user may start with a broad query and continuously refine it based on the results generated from the search algorithm. Eventually, the user exits the system, which is modeled by the terminal state. The user can make a purchase at any point during the session.

$(P, r)$ and $\tilde{M} = (\tilde{P}, \tilde{r})$ can be quite different. In fact, comparing these two estimated models provides an idea of the existing noise in the estimated models. For example, the average $L^1$ norm of the rows of $P^{ranker1} - \tilde{P}^{ranker1}$ (also $P^{ranker2} - \tilde{P}^{ranker2}$) is about $0.16$, suggesting that the average total variational distance between transition probability vectors of $P$ and $\tilde{P}$ is $0.08$, which is empirically quite significant.

For comparison purposes, we include the performance of several baselines defined below:

- unregularized MDP policy: this policy is optimal for model $M$ and is applied to $\tilde{M}$ without any regularization.

- one-shot policy (OSP): this policy selects action ranker 1 for a particular keyword $s$ if immediate reward from ranker 1 is larger, i.e., $r_s^{ranker1} > r_s^{ranker2}$.

- regularized one-shot policy (OSP$_\lambda$): this policy selects ranker 1 for a particular keyword $s$ only if $r_s^{ranker1} - \lambda > r_s^{ranker2}$.

Figure 8 shows the performance of $L^1$-regularized MDP policy, the unregularized MDP policy, OSP, and OSP$_\lambda$ on

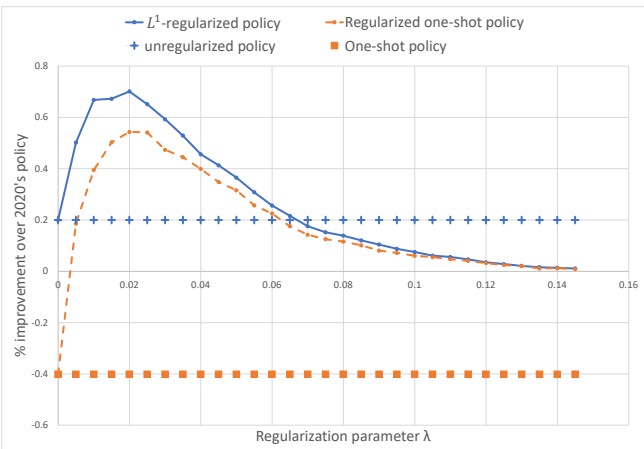

Figure 8: Performance comparison on an online shopping store dataset: The $L^1$-regularized policy accounts for the prior information through $\lambda$. As $\lambda$ increases, the $L^1$-regularized policy favors action 0 over action 1. The incorporated prior allows the $L^1$-regularized MDP policy to outperform the unregularized MDP policy learned from $M$. Since the MDP based approach accounts for the delayed rewards by modeling the session interaction, it outperforms the regularized one-shot policy $\text{OSP}_\lambda$ for all values of $\lambda$.

model $\tilde{M}$, where all these policies are learned from model $M$. We observe that (a) the $L^1$-regularized MDP policy shows about $0.7\%$ improvement over the ranker 2 policy, and the improvement over the ranker 1 policy is in the range of 21-22%, (b) the $L^1$-regularized policy outperforms both OSP and $\text{OSP}_\lambda$, suggesting that the MDP model is beneficial as the OSP and $\text{OSP}_\lambda$ do not factor in delayed rewards, and (c) the $L^1$-regularized policy outperforms the unregularized MDP policy by accounting for prior information in the form of regularization parameter $\lambda$. The best performance is obtained with $\lambda = 0.02$. As the value of $\lambda$ increases, the regularized policies prefer ranker 2 over ranker 1. When $\lambda$ is increased further, the learned policy ends up selecting ranker 2 for all the states. This is why its performance becomes similar to ranker 2 in Figure 8 for large values of $\lambda$.

The same experiment is repeated for the RE-regularized policy and we observe a performance similar to that of the $L^1-$regularized policy with the best improvement of $0.69\%$ coming at $\kappa = 0.001$ and $q_s^{ranker1} = 10^{-8}$.

The above experiments suggest that hyperparameters $\lambda = 0.02$ and $(\kappa = 0.001, q_s^{ranker1} = 10^{-8})$ perform the best for the $L^1$-regularized policy and the RE-regularized policy, respectively. To validate our results, we performed another experiment with these hyperparameters, where we learned the model $M$ with a new action space $A = \{ranker2, ranker3\}$ by collecting user logs for two different non-overlapping time periods in which ranker 2 and ranker 3 were deployed. In this scenario, ranker 3 was more recently deployed relative to ranker 2. The test data is also generated by collecting user logs on a hold-out time period where ranker 2 and ranker 3 were previously deployed. In

this situation, the prior is that the ranker 3 is on average better than ranker 2 and it has been incorporated in the computation of the $L^1$-regularized and RE-regularized policies. The results are reported below in Table 1.

| Algorithm | % improvement over ranker 3 |
|---|---|
| unregularized policy | $-0.1$ |
| $L^1$-**regularized policy** | **0.214** |
| **RE-regularized policy** | **0.207** |
| One-shot Policy | $-6.33$ |
| Regularized one-shot Policy | $-2.92$ |

Table 1: Performance comparison of different policies with pre-learned hyperparameters. The task is to identify the better search algorithm among ranker 2 and ranker 3 for a given query. The $L^1$-regularized and RE-regularized policies outperform other approaches as (a) they account for delayed rewards through MDP based session modeling and (b) they are robust to noise by factoring in the prior knowledge.

We make several observations from the results listed in Table 1. First, there is a significant difference between the performance of the regularized one-shot policy and the regularized MDP policies, as the sessions were of longer range in the collected data. For example, a typical query improved using older search algorithm is "mens gifts". As this query leads to sessions of larger length on the shopping store, the regularized MDP policies allow us to factor in the delayed rewards and suggest the appropriate search algorithm for the query. An empirical approach such as $\text{OSP}_\lambda$ fails to evaluate the quality of search results in this case as it focuses only on the immediate rewards. Second, the regularized MDP policies outperform both ranker 2 and ranker 3, whereas all other policies are worse of than the simple strategy of selecting ranker 3 for all queries. This is because the $L^1$-regularized and RE-regularized MDP policies account for the noise present in the model and incorporate the known prior information appropriately.

## 5    Conclusions

In this paper, we study the problem of learning policies where the parameters of the underlying MDP $\bar{M}$ are not known but instead estimated from empirical data. Simply learning policies on the estimated model $M$ may lead to poor generalization performance on the underlying MDP $\bar{M}$. To address this issue, we propose a Bayesian approach, which regularizes the objective function of the MDP to learn policies that are robust to noise. Our learned policies are based on $L^1$ norm regularization and relative entropic regularization on the objective function of MDP. We show that our proposed regularized MDP approaches end up penalizing the reward of less preferred actions, thereby giving preference to certain actions in $A$ over others.

To validate the performance of proposed algorithms, we evaluate the performance on both synthetic examples and on the logs of real-world online shopping store data set. We demonstrate that a policy learned optimally on $M$ without

any regularization can even do worse than a simple policy that always selects one of the actions $a \in A$ for all the states. Our experiments reveal that the un-regularized policies are not robust to noise in probability and reward tensors. On the other hand, the regularized MDP policies significantly outperform other baseline algorithms both on synthetic and real-world numerical experiments.

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
