# OpenReview forum: "Bayesian regularization of empirical MDPs"
_ICLR.cc/2023/Workshop/RRL — RRL 2023 Poster_

### Official Review · Reviewer_inuP · 2023-02-20
**Type of policy transfer method. Could use more experimental evaluations. Well written.**

**Rating:** 3
**Confidence:** 3

**Review:**


Review Summary
--------------

Recommendation: poster

__Relevance for workshop: 7/10__

Type of policy transfer that can be used to bias a policy in a new target task towards a set of known optimal actions.
Mainly discussed for specific scenarios where a policy is learned from a noisy approximated MDP model.

__Scientific quality: 6/10__

Well-derived methods, but I am missing some simple baseline approach and a comparison to other policy transfer methods.

__Paper quality: 7/10__

Well written and clear. Some details about figures could be improved.

(Points:  lowest: 0/10 means, highest: 10/10, >=5 means a recommendation to be included to the workshop)


Paper Summary
-------------

The paper introduces two methods to learn robust policies for imprecise (noisy) MDP models (transition and reward function).
Both are based on a Bayesian approach taking into account prior knowledge about a set of actions that are preferred over others per state.
L1 regularization and entropy regularization are introduced to incorporate this knowledge increasing the preference for these actions in the learned policy of the noisy model.
Methods are evaluated on two simulated examples and an online shop recommendation task where an agent selects a search result ranking method.

The paper is well-written and understandable.
Unfortunately, I did not have the time to follow all your derivations in detail, but your equations and methods seem sound.
I am mainly missing some experimental work and a comparison to other methods. See below.


Major Points
------------

1) Missing baseline approaches.
It would be good to compare your method to other baselines that have a similar assumption, i.e. having a set of preferred actions $\xi(s)$.
I am not an expert in this particular field and can not recommend certain other methods from the literature, but I encourage you to look for them.
But a simple method to introduce a bias for a preferred set of actions $\xi(s)$ over others is to simply adjust the policy probability by adding a bonus that can be parameterized.
Given your policy $\\pi$ learned for $\\bar{M}$ we can construct a biased policy $\pi'$ with:
$\\pi'(s,a) = \\frac{\pi(s,a) + \\beta (a \in \\xi(s)?)} {\\sum_a (\\pi(s,a) + \\beta  (a \in \\xi(s)?)}$, where $\\beta \\in [0, \\infty]$, and
$(a \in \\xi(s)?)$ is the identify function which is 1 if $a \in \\xi(s)$, otherwise 0.
It would be good if you could compare at least to this simple method and discuss the possible advantages, and disadvantages.

2) Evaluation of $\\kappa$.
How did you choose $\\kappa$ for the results in Fig 3?
I recommend doing an evaluation over both $\\kappa$ and $q_s^1$. You could show the results in a 2D figure where the color encodes the performance.
(Example: https://www.exobrain.online/2016/08/09/matplotlib-pyplot-imshow/)

3) Why this particular simulated MDP?
It would be great to explain why you choose this particular form of simulated MDP (section 3).
What are the characteristics that make it a good example to evaluate your approaches?
For example, what can it especially highlight. Maybe also a discussion of what it can not highlight.

4) Evaluation: The number of actions is always 2.
In all your tasks, the number of actions is always 2, but your method should work with a larger set of actions.
Thus, you should include some evaluations in settings with more actions.

5) Related work in policy transfer.
Your approach is closely related to approaches that do policy transfer as your prior knowledge of $\\xi(s)$ can be interpreted as a policy.
I would suggest discussing this connection and differences in your related work section.
See chapter 4.3 of Zhu, Zhuangdi, Kaixiang Lin, and Jiayu Zhou. "Transfer learning in deep reinforcement learning: A survey." arXiv:2009.07888 (2020).

6) No code available.
I recommend the authors to publish their code (at least for the simulated example).
This helps to ensure reproducibility.


Minor Points
------------
1) page 1, last paragraph: " ... than the naive policy $\pi$ from M when ... " : Do you mean here instead $\bar{M}$, the approximated MDP model
2) the font size of the legends in your figures is inconsistent between figures
3) the color of the lines in your figures could be more consistent. Use one color for every algo (e.g.: blue - L1, green - regularized $\pi$, orange - standard $\pi$).
  You use orange for standard $\pi$ in most figures but then also orange in Fig 8 for the regularized $\pi$.
4) Table 1: a) Not centered. b) You can remove the hline's between the algos (rows) to look nicer.
5) Fig 7: font size of text elements could be larger
6) Fig 6: x-label is missing